# How Do Patients Understand Questions about Lower Urinary Tract Symptoms? A Qualitative Study of Problems in Completing Urological Questionnaires

**DOI:** 10.3390/ijerph19159650

**Published:** 2022-08-05

**Authors:** Florine W. M. Schlatmann, Michael R. van Balken, Andrea F. de Winter, Igle-Jan de Jong, Carel J. M. Jansen

**Affiliations:** 1Department of Urology, University Medical Center Groningen, 9700 RB Groningen, The Netherlands; 2Department of Urology, Rijnstate Hospital, 6815 AD Arnhem, The Netherlands; 3Department of Health Literacy and Prevention, Health Science, University Medical Center Groningen, 9700 RB Groningen, The Netherlands; 4Department of Communication and Information Studies, University of Groningen, 9712 EK Groningen, The Netherlands; 5Language Centre, Stellenbosch University, Stellenbosch 7600, South Africa

**Keywords:** lower urinary tract symptoms questionnaires, health literacy, working-aloud method

## Abstract

Lower urinary tract symptoms are common complaints in ageing people. For a urological evaluation of such complaints in men, the International Prostate Symptom Score (IPSS) is used worldwide. Previous quantitative studies have revealed serious problems in completing this questionnaire. In order to gain insight into the nature and causes of these problems, we conducted a qualitative study. Not only the purely verbal IPSS was studied but also two alternatives, including pictograms: the Visual Prostate Symptom Score (VPSS) and the Score Visuel Prostatique en Image (SVPI). Men aged 40 years and over with an inadequate level of health literacy (IHL; *n* = 18) or an adequate level of health literacy (AHL; *n* = 47) participated. Each participant filled out one of the three questionnaires while thinking aloud. The analysis of their utterances revealed problems in both health literacy groups with form-filling tasks and subtasks for all three questionnaires. Most noticeable were the problems with the IPSS; the terminology and layout of this form led to difficulties. In the VPSS and SVPI, the pictograms sometimes raised problems. As in previous research on form-filling behavior, an overestimation by form designers of form fillers’ knowledge and skills seems to be an important explanation for the problems observed.

## 1. Introduction

In urology, lower urinary tract symptoms (LUTS) are common complaints associated with ageing. LUTS may cause great discomfort and reduce the quality of life. For urologists, a thorough evaluation of a patient’s complaints is needed to gain insight of the symptoms and determine the management of LUTS. Such an evaluation usually includes a combination of medical history, examination, flow rate measurement and bladder diaries, but the use of a validated questionnaire is also strongly recommended in the guidelines of the European Association of Urology as well as in the guidelines of the American Urological Association [1,2].

### 1.1. The International Prostate Symptom Score (IPSS)

Worldwide, the most frequently used questionnaire for LUTS in men is the *International Prostate Symptom Score* (IPSS). The IPSS is intended to convert subjective symptoms into quantitative values. It can be used for assessing symptom severity before, during and after treatment. The questionnaire consists of seven questions to evaluate LUTS. This set of questions is also known as the *AUA Symptom Score Questionnaire* (AUA-SS or AUA-7). Questions 1, 3, 5 and 6 refer to storage symptoms; questions 2, 4 and 7 refer to irritative symptoms. The answers to the first seven questions range from 0 (never a problem) to 5 (almost always a problem). A total score of 0 to 7 on these seven questions indicates mild symptoms, 8 to 19 indicates moderate symptoms, and 20 to 35 indicates severe symptoms. The last question is about quality of life in relation to urinary symptoms (utQOL). Here, scores range from 0 to 6, with a high score indicating a low utQOL [3,4].

The version of the IPSS applied in this study has been used for many years in a number of Dutch hospitals, including the Rijnstate Hospital in Arnhem where this study was conducted. The translations from English follow the wording in the original version introduced by Barry et al. (1992, S195) in the *Journal of Urology* [5]. The layout bears similarity to the version of the IPSS presented by Burnett and Wein (2006, S20) in the same journal [6] (see also Ref. [7]). See Figure 1.

Several studies have revealed problems with the completion of this entirely verbal questionnaire [3]. Self-administration of the IPSS was found to lead to incomplete questionnaires in 53–73% [4,8,9]. The level of education appears to be related to the level of completion of a self-administered IPSS. The lower the patients’ level of education, the smaller the chance that they are able to complete the IPSS unaided [10,11,12]. A similar relationship was found between the level of education and correctness of symptom representation when completing the IPSS [13,14]. A grade 6 reading level is required to read and understand this questionnaire and due to reading and comprehension problems, a substantial group of patients needed help from others in completing the IPSS [15].

Although the IPSS is often regarded in urology as the gold standard reference for the evaluation of LUTS by patients [16,17], these results obviously pose a serious threat to the validity and thus the usefulness of this questionnaire. Johnson et al. (2007, p. 626) suggest that a questionnaire that provides urologists with inaccurate information because patients did not understand the questions may lead to serious problems: overtreatment in the form of unnecessary procedures, surgery or medication, or undertreatment in the form of patients not receiving the necessary therapy for unreported symptoms [13].

### 1.2. The Visual Prostate Symptom Score (VPSS)

In view of the limitations of the IPSS [10,14,15], a group of South African urologists designed a simplified, visual alternative to assess the strength of the urine stream for patients who are unable to read and write. The questionnaire is called the *Visual Prostate Symptom Score* (VPSS) [9,18,19]. The VPSS consists of three visual questions that assess urinary symptoms (urinary frequency during day and night, and strength of the urine stream), plus a visual scale to rate the utQOL. There are no verbal explanations in the VPSS, except “or more” in the last answer option of the two frequency questions. A score of 1 to 6 for the first three questions indicates mild symptoms, 7 to 13 indicates moderate symptoms, and 14 to 17 indicates severe symptoms [17]. See Figure 2.

A number of studies were conducted to evaluate the VPSS, often comparing scores on corresponding questions of the VPSS and the IPSS and also comparing total VPSS scores to total IPSS scores (for an overview, see Appendix A, partly based on Ref. [3]). Given the shortcomings of the IPSS (see above), the value of such comparisons may be questioned, however. If a problematic measuring instrument A and a new measuring instrument B turn out to give similar results, this outcome can hardly be considered as support for the quality of instrument B, and if the results are different, that does not automatically imply that the measurements with instrument B are correct. To circumvent this problem, several studies also calculated the correlations between IPSS and VPSS scores on the one hand and objectively determined uroflowmetry parameters on the other. Comparisons with Qmax (maximum strength of the urine stream) and Qave (average strength of the urine stream) reveal that the VPSS usually performed somewhat better than the IPSS (see Appendix A). However, it should be noted that the strongest correlation (*r*) that was found (between VPSS score and Qmax) was −.729, which implies that never more than 53.14% of the variance in flowmetry outcomes (*R*^2^) could be explained by either IPSS or VPSS scores.

Two studies compared how many patients were able to complete the IPSS or the VPSS without assistance. One study found results in favor of the VPSS [9], the other study in favor of the IPSS [17]. In both studies, it was found that more participants with a high level of education than with a low level of education were able to complete each questionnaire. The majority of the less educated participants felt that the IPSS was easier to understand than the VPSS [17], while a small majority of the highly educated patients felt that the VPSS was easier to understand [9]. Another study found that some of the pictograms were difficult to interpret due to their small size and the low contrast with the background. Here, a substantial group of participants experienced symptoms of urgency that were not referred to in the VPSS and that they hence could not report [20].

### 1.3. The Score Visuel Prostatique en Image (SVPI)

In France, a group of urologists also designed a visual questionnaire as an alternative to the IPSS. The questionnaire they developed is called the *Score Visuel Prostatique en Image* (SVPI), translated into English as the “Visual Prostatic Symptom Score” [21]. The SVPI consists of the three questions that are also included in the South African VPSS for assessing urinary symptoms (SVPI questions 1, 2, 4 corresponding to VPSS questions 1, 2, 3, respectively), plus an extra question (SVPI question 3) about urgency. Just like the IPSS and VPSS, the SVPI ends with a question that assesses the quality of life in relation to the symptoms of the urinary tract (SVPI question 5). In the SVPI, each question contains a start pictogram followed by a number of answer options, also in the form of pictograms. Each question is also asked in words. Possible scores for the questions are as follows: question 1 and question 2: 0–6; question 3: 0–5; question 4: 0–4; question 5: 0–7. In publications introducing the SVPI, threshold values are not mentioned [21,22]. See Figure 3.

Two validation studies found statistically significant correlations between SVPI scores and IPSS scores. No correlations with objective flowmetry measurements are reported, however, and no information is provided on the health literacy status or educational level of the patients. The authors’ claim that the SVPI is a simple questionnaire is supported by opinions of urologists and GPs but not by data directly provided from patients who participated in the studies [21,22].

### 1.4. Research Questions

The research carried out so far on the validity of both the IPSS and its visual alternatives, the VPSS and the SVPI, has been mainly quantitative in nature. The aggregate results provide useful information about the frequency of problems experienced by participants in completing the questionnaires. However, it is unclear what the nature of the problems the patients were facing was, and what caused these problems. Which tasks and subtasks that the fillers of these forms have to perform lead to difficulties and to which characteristics of the questionnaires in relation to the behavior of the form fillers can these problems be attributed? Moreover, it is unclear whether there is a relationship between the level of literacy of patients on the one hand and the type of completion problems and underlying causes on the other. Clarity on these issues could provide a sound basis for a new questionnaire that is less difficult to complete for different groups of patients and that thus would improve the quality of information for treating urologists. The quantitative studies performed to date do not yet provide sufficient insight into how the patient perceives, interprets and reacts to the items in the LUTS questionnaires and how this can lead to the patient not completing the questionnaire correctly. As pointed out in Ref. [23], for instance, such completion errors can have a negative effect on the validity of the information (p. 1392).

Given this state of affairs, we decided to conduct a qualitative study among Dutch men, aimed at gaining more insight into the nature of the problems with existing LUTS questionnaires and the underlying causes for patients with various levels of health literacy. We also aimed to relate the outcomes of this qualitative study to the results of previous studies. The research questions for the present study were as follows:What kind of problems do men face when trying to complete an existing LUTS questionnaire?What causes can be identified for these problems?What is the relationship between these problems and the causes thereof on the one hand, and the level of health literacy of the participants on the other hand?

## 2. Materials and Methods

### 2.1. Design

We performed a qualitative study using the working-aloud method developed by Jansen and Steehouder [24] (see Section 2.5). Each of the randomly distributed LUTS questionnaires was completed while thinking aloud by at least 5 participants with an inadequate level of health literacy (IHL) and at least 10 participants with an adequate level of health literacy (AHL). All participants filled out one questionnaire, after which a short interview was held. To measure each participant’s level of health literacy, the *Set of Brief Screening Questions* (SBSQ) [25] was used (see Section 2.6).

### 2.2. Participants and Setting

Men of 40 years and older who had not previously filled out IPSS, VPSS or SVPI were invited to participate in the study. Pre-specified minimum numbers per questionnaire were 15 men with AHL and 5 men with IHL. In all conditions, new participants were invited until no more data were found that provided new insights (saturation).

To find suitable participants who were willing to complete the IPSS, the VPSS or the SVPI, initially, a convenience sampling method was applied, with a face-to-face approach of men visiting an outpatient urological clinic in the Dutch city of Groningen. Not all men who were invited to participate suffered from LUTS; outpatient urological clinics are also visited by people who have other complaints.

This first phase of data collection yielded 48 participants, including 2 patients with IHL. All 48 participants filled out one of the LUTS questionnaires in the hospital, in the presence of the primary investigator.

In view of the small number of IHL participants, a consecutive sampling method was applied. In cooperation with social service organizations, possible additional male participants with IHL without urology problems were approached face-to-face by their literacy trainer. The part of the study with these participants took place at the social service organizations, again, in the presence of the primary investigator. In this way, 17 additional participants could be included, one of whom with AHL. The questionnaires were distributed in a fixed order: participant 1 received the IPSS, participant 2 the VPSS, participant 3 the SVPI, participant 4 the IPSS, participant 5 the VPSS, and so on. For an overview of patient characteristics and the distribution of the patients among the three questionnaires, see Table 1.

### 2.3. Procedure

After a standardized introduction about the aims of this study (see Appendix B), first, the level of health literacy was assessed for each participant using the SBSQ (see Section 2.6). From the answers to the SBSQ, the test leader could deduce the health literacy status of the participant, but care was taken to ensure that this did not affect the continuation of the session. During the test administration, a predetermined code of conduct was followed; the test leader’s utterances to encourage thinking aloud were standardized, and no assistance was provided to either AHL or IHL participants.

After this, the participant completed one of the LUTS questionnaires while thinking aloud. For this task, the primary investigator instructed each participant as follows: “Please say out loud what you read and think as you complete this questionnaire. Please act as if you were alone in your room and talking to yourself”. Participants were told not to worry about whether or not the researcher would be able to understand what they were saying. They were told that, when silent, after a while, they would be reminded to think aloud. It was further pointed out that it was not the role of the researcher to provide assistance. When the participant asked a question, the standardized answers from the primary investigator were: “What would you do if I wasn’t here?” or “Please do what you think is best”. For reassurance, it was emphasized that the participant was not the subject of the test, but the questionnaire was. If participants indicated that they would be more comfortable in the presence of their partner or coach during the study, their presence was allowed under the condition of no help. There was no time limitation.

Immediately after filling out the questionnaire, a retrospective interview was conducted. The questionnaire was discussed question by question, and the participant was asked to indicate ambiguities in the questionnaire and possibilities for improvement. After going through the questionnaire, six standardized questions were asked:What did you think of the questionnaire?Is the layout of the questionnaire clear, and if not, what could be improved?Is the questionnaire more difficult or easier compared to other forms you are familiar with?Did you miss any general information, such as an introduction at the top of the questionnaire?How did you experience this test?Should you have completed this questionnaire for yourself at home, would you have proceeded in the same way as you have done now?

### 2.4. LUTS Questionnaires

Included in this study were the IPSS (see Section 1.1) and two of its mainly visual alternatives: the VPSS, developed in South Africa (see Section 1.2) and the SVPI, developed in France (see Section 1.3).

Dutch versions of all three existing LUTS questionnaires were used. The Dutch version of the IPSS was provided by Rijnstate Hospital in Arnhem. For the Dutch version of the VPSS, only two words had to be translated. In the response options to the first two questions, the frequency indication (“6 of meer”) was used, which is Dutch for “6 or more”. In the SVPI, all verbal phrases were translated from French into Dutch, following standardized guidelines [26]. Three professional translators at the Language Centre of the University of Groningen who had Dutch as their native language independently forward-translated the SVPI. After consultation of the primary investigator with the second investigator and the main investigator, differences in the translations were discussed in a meeting of the translators and the primary investigator. It was decided that some minor textual changes in translations had to be made. The final version was backward-translated by a professional translator of the Language Centre of the University of Groningen who had French as his native language. No differences compared with the original SVPI were found. Below, the original versions of the IPSS, the VPSS and the SVPI are presented.

### 2.5. Working-Aloud Method

In previous research on form-filling problems and their causes, the working-aloud-work method proved to be a fruitful approach. This method was introduced by Jansen and Steehouder in their research on government forms [24] (pp. 42–45). Like they did, we asked participants to think aloud while filling out a form. We also used three other techniques (product analysis, behavioral observation and retrospective interviews) to collect information about the problems people face when completing a form, in our case, a LUTS questionnaire. None of these techniques guarantees a complete representation of the cognitive processes about which the researcher may be concerned. However, each technique can make a valuable contribution to an overall understanding of form-filling problems and their causes [24] (pp. 41–45), [27,28,29].

#### 2.5.1. Techniques

One of the techniques used in the working-aloud method is product analysis. The researcher evaluates the product, which is the result of the task performance (here: a completed form), and tries to understand the way in which the product came about. Product analysis is important to identify problems, but it is insufficient to explain the occurrence of these problems. The participant may have arrived at the same result in different ways. For this reason, additional techniques are needed.

Another technique is to record the participant’s behavior. Behavioral observation can be important for tasks in which material actions play a role. However, when mental action is involved, only indirect observation is possible—in this case, of the moments when the participant picks up and puts down the pen, goes from one question to another, has a certain facial expression, and so on.

A third technique is to conduct a retrospective interview. After the task has been completed, the researcher asks the participant for information about the cognitive processes involved in the task performance. The advantage is that the interviewer can obtain clarifying information about specific steps in the process. A disadvantage is that the participant may not remember exactly how he or she proceeded. There is also the risk of retrospective rationalization. What actually happened is influenced by the participant’s ideas about what should have happened.

The technique that lies at the heart of the working-aloud method is to record the participants’ thoughts as they verbalize them during the performance of their task [24,27,28,29,30,31,32]. Asking participants to think aloud while performing their task and using the utterances as research data is often the most direct technique available. The data are collected synchronously with the process; the researcher does not intervene in the process; rationalization by the participant is unlikely with correct instruction; and the method does not influence the problem-solving process. However, there will always be some level of incompleteness. Not everything the participant thinks will be verbalized. Participants often report that their utterances are incomplete, and it must often be assumed that they had intermediate thoughts that were not verbalized [33] (p. 33).

In this study, we first asked each participant to fill out one of the LUTS questionnaires while thinking aloud. Immediately after the participant did so, a retrospective interview was conducted. The participant was asked to describe how he had performed his task, and specific questions were asked to gain a better understanding of how he had dealt with problems that had arisen. As a result, four types of data were available: *product data* (completed questionnaires), *behavioral observations* (for example, regarding the order of answering questions), *thinking aloud data* (verbalizations) and *problems experienced by the participants* as reported during the retrospective interviews.

#### 2.5.2. Task Analysis

In order to obtain a clear picture of the problems in completing a form and their causes, it is useful to dispose of a task analysis: an overview of the tasks and subtasks that the form filler has to perform. With such a task analysis, it is possible to determine with which of these tasks and subtasks the filler does or does not experience problems. Causes can then be sought for the problems met when carrying out various tasks and subtasks. In the analyses performed in this study, we distinguished three levels of tasks, just as Jansen and Steehouder [27] (p. 183) did in their study on problems in filling out government forms: control tasks, interpretation tasks and functional tasks. See Figure 4.

First, form fillers need to understand the textual and the visual information in the questionnaire. To be able to do this, they must perform *interpretation tasks*. First of all, they have to understand the textual and the visual elements, each separately and in their interrelationship: the *semantic* aspect. From the meaning of the textual and the visual elements, form fillers must also deduce which functional actions they must perform to complete the form: the *pragmatic* aspect.

Then, there are tasks that belong directly to the actual task: *functional tasks*. Here, three subtasks can be distinguished:*generating data*: producing the requested information;*verifying*: checking whether an item of information fits into a particular category;*transforming*: converting an item of information into a code, such as a graphic sign (check a box, for instance).

*Monitor tasks* include the subtasks that control the process of which the functional and interpretive tasks are a part. Three types of monitor subtasks can be distinguished:*orientation*: obtaining an overview of the document and its various parts, and of the overall task and the various steps that it comprises;*selection*: deciding which functional tasks and which interpretation tasks should or should not be carried out given the form filler’s individual situation;*checking*: ascertaining whether the task performance is proceeding correctly and making corrections where necessary;*switching*: interrupting the performance of interpretation tasks to carry out functional tasks and vice versa.

We speak of a problem whenever the form filler has apparently wrongly failed to perform one of the above-mentioned subtasks, performed a subtask incorrectly or performed a subtask correctly but based on incorrect reasoning [27]. Whether or not a problem had occurred was inferred from the completed forms, from the behavioral observations, from the thinking-aloud data, from the retrospective interviews or from a combination of these.

### 2.6. Measuring Level of Health Literacy: SBSQ

To measure the participant’s level of health literacy, the SBSQ, consisting of three questions, was used. The SBSQ was developed and tested by Chew et al. (2004) [25]. As the authors note, until 2004, no practical method for identifying patients with an inadequate level of heath literacy existed. A limited set of questions was composed for identifying patients with inadequate or marginal health literacy. Based on a comparison with the outcomes of a validated health literacy measure, the Short Test of Functional Health Literacy in Adults (S-TOHFLA, comprising 40 items), the following questions were found to be effective in detecting an adequate or inadequate level of health literacy: “How often do you have someone help you read hospital materials?”; “How confident are you filling out medical forms by yourself?”; and “How often do you have problems learning about your medical condition because of difficulty understanding written information?”. The answer options for each question are 0–4. In 2008, Chew et al. [34] performed a large-scale validation study in a veteran population in the USA (*n* = 1.796). The score ≤2 provided optimum sensitivity and specificity [25,34]; this score is considered as an optimal screening threshold in most studies [35] (p. 136).

### 2.7. Data Collection and Analysis

All SBSQ questionnaires and LUTS questionnaires completed by participants were collected. A mean score of ≤2 was interpreted as indicating inadequate health literacy (IHL) and a score >2 as indicating adequate health literacy (AHL). Furthermore, field notes were made during each working-aloud session, including the interview. Audio recordings of the introduction, the thinking-aloud utterances and the interview were also produced for further analysis by the principal investigator, in six cases together with the second investigator (see Section 2.8).

For analyzing the data, a scoring system was used that was developed by Jansen and Steehouder [24]. The problems that the participants from the IHL group or the AHL group encountered while performing the various tasks and subtasks in Figure 4 were scored as: “Question answered correctly, no problem occurred”; “Question answered correctly but reasoning not completely correct” (based on the information expressed by the participant while thinking aloud); “Question not answered correctly; answer not acceptable”; and “Question not answered”. Additionally, following Ref. [24], possible causes were categorized as *lack of prior knowledge*, *lack of skills* or *lack of willingness* to work through the form.

Note that there were also cases where none of the problem categories applied, but comments were provided that indicated some confusion. For example, the flashing light in the third question of the SVPI initially caused confusion among some participants. However, this icon did not cause any problems as defined above.

### 2.8. Characteristics of the Investigators

The primary investigator and first author (FS; female, MD, resident of urology) acted as the test leader. She conducted the interviews and coded the data as described for all participants. Thinking-aloud utterances and answers in the interviews of two randomly chosen participants per questionnaire were analyzed by the primary investigator together with the second investigator (MvB). In one case, there was a minor doubt in the scoring of a thinking-aloud-utterance, which was cleared up after inspection of the field notes. Prior to the study, the primary investigator was trained in interviewing and scoring by the principal investigator of this study (CJ). No medical or other relationship between the investigators and the participants was established prior to, during or after the study.

## 3. Results

Table 1 presents the characteristics of the participants.

Below, we first present the problems that participants apparently experienced with interpretation tasks (Section 3.1), then, problems with functional tasks (Section 3.2), and finally, problems with monitor tasks (Section 3.3). For each of these problem types, we first discuss what happened when participants filled out the IPSS, then, what problems were encountered when the VPSS was filled out, and finally, what happened when the SVPI was filled out. Space does not permit presenting all the available details here. A quantitative overview can be found in Appendix C; complete working-aloud reports can be requested from the first author.

Here, we concentrate on the problems that may threaten the validity of a urologist’s interpretation of the answers recorded on the questionnaire. Where relevant, we also refer to possible causes of the problems, such as a term, a pictogram or a question that calls on prior knowledge or skills that some of the form fillers apparently do not possess. Both when discussing problems and possible causes, we distinguish between AHL and IHL participants. We illustrate our findings with one or more quotations from the participants; all quotations were translated from Dutch into English by the first author. Finally, we discuss the participants’ answers to standardized questions from the retrospective interviews (Section 3.4).

### 3.1. Problems with Interpretation Tasks

#### 3.1.1. IPSS

The texts presented *semantic interpretation* problems at different levels. Four out of seven IHL participants, including two participants with a low command of Dutch, had problems with the terms “bladder”, “weak”, “urine stream” and “month”. Three out of seven IHL participants did not answer one or more questions. At the interviews, these participants explained why; they simply had not understood these questions. The interpretation of the answer options also regularly posed problems, with the response option “less than 1 out of 5 times” described as unclear by both IHL and AHL participants while thinking aloud and also at the interviews. For example, an IHL participant remarked during the interview:

“First thing I think: So did I only pee 5 times in a month? Or 5 times a day?”

*Pragmatic interpretation* problems occurred in 4 out of 16 AHL participants and in 4 out of 7 IHL participants when trying to interpret what was meant by “over the past month”, as they noted during the interviews. In addition, the table design of the IPSS often presented pragmatic interpretation problems. For 3 out of 16 AHL participants, it was difficult to relate the questions in the rows to the answer options in the columns, as they explained in the interviews.

An AHL participants said:

“In the beginning it was difficult to connect the X and Y axes”.

An IHL participant commented:

“The answer options are tricky. In the beginning, it seemed that the answer most to the left meant ‘never’ and the one most to the right meant ‘almost always’. Then when my reaction to a question was ‘pretty good’, I filled out the box next to ‘almost always’.”

From the thinking-aloud utterances, it came to light that three out of seven IHL participants and one AHL participant used the numbers in the answer boxes without trying to interpret the answer options presented in the top row. This problem was confirmed during the retrospective interview. Surprisingly perhaps, the reversed order of the answer options in the last question (utQOL) did not seem to raise specific problems.

#### 3.1.2. VPSS

The pictograms in the VPSS posed many *semantic interpretation* problems for both AHL and IHL participants. For 5 out of 15 AHL participants, what was being asked in the first question did not become clear until they had seen the second question, as became evident from the thinking-aloud utterances. One out of fifteen AHL participants did not understand that the number of urinating puppets visualizes the frequency in the first and second question. Instead, he tried to assess the strength of the urine stream of these puppets, as he reported in the interview. For three out of six IHL participants, it was unclear what was being asked in question 1, even after seeing question 2. As a result, these participants did not answer the first two questions.

Additionally, different interpretations of the figure before the first question arose: a man apparently doing groundwork. An AHL participant said during the interview:

“I thought it was about how often I pee during my work.”

Another AHL participant noted during the interview:

“I filled out the frequency during the morning because I always work in the garden in the morning only.”

One out of six IHL participants said in the retrospective interview that he found the third question unclear, but from his explanation in the retrospective interview, it seemed that he had answered this question correctly. From the thinking-aloud utterances of 7 out of 15 AHL participants and 4 out of 6 IHL participants, it appears that the fourth question was unclear to them. As a result, a correct answer was apparently based on incorrect reasoning five times (five out of fifteen AHL participants), three answers were incorrect (one out of AHL participants and two out of six IHL participants), and in two cases, there was no answer at all (two out of six IHL participants). One out of fifteen AHL participants ticked two answers in the fourth question, despite his thinking-aloud utterances showing that he had interpreted the question correctly.

*Pragmatic interpretation* problems were noted for several participants. In particular, the lack of a concrete question at the bottom of the form about quality of life led to uncertainty for men in both literacy groups about what they were expected to do. For example, 1 out of 15 AHL participants reported during the interview:

“I miss the question or text, I don’t really know what is expected of me.”

#### 3.1.3. SVPI

The first and the second question in the SVPI, about daytime frequency and nighttime frequency, respectively, did not seem to present *semantic interpretation* problems. One out of five IHL participants (with a low command of Dutch) had problems understanding the Dutch term (“hoge nood”) for “envies pressantes” (pressing urges) in the third question. As a consequence, he did not complete this question.

The third question posed *semantic interpretation* problems for 12 out of 16 AHL participants and 2 out of 5 IHL participants, the interviews revealed. Because of the red flashing light preceding this question, placed to the left of the answer option for least urgency, confusion in the interpretation of the answer options arose. There was also confusion about the first answer option, which was caused by the red diagonal in the picture. This confusion was noted in the interviews in 7 out of 16 AHL participants and 2 out of 5 HL participants.

For two out of sixteen AHL participants, a *pragmatic interpretation* problem occurred. They noted that they had excluded the first response option in the third question because of this red line; as a result, they felt they had to choose from fewer response options. An AHL participant said during the interview:

“The flashing light is on the left, while the answer on the left actually indicates ‘no need’. Also, the answer option on the left with the red stripe represents ‘emergency’ to me, while this is precisely the answer without emergency. Red stands for alarm!”

One out of sixteen AHL participants and one out of five IHL participants recommended not only removing the red line but also changing the answer option for least urgency to a stationary puppet.

*Semantic interpretation* problems also arose with the fourth question. The thinking-aloud utterances of 3 out of 16 AHL participants revealed that, for them, it was confusing and unnatural that the highest number is under the weakest urinary stream. During the retrospective interview, an AHL participant noted:

“You would think the stronger the stream the higher the number. That made me doubt for a moment.”

Additionally, this participant had problems in interpreting the Dutch translation (“kracht van de straal”) of the French expression “force du jet” (strength of the urine stream) in the fourth question. However, according to the thinking-aloud utterances, he succeeded in answering this question correctly with the help of the pictogram, as was verified during the retrospective interview. Three out of five IHL participants felt the figure in the fourth question was too small. This made it difficult for them to see differences between the different strength levels of the urine stream, they noted during the interviews. In the retrospective interviews, two out of five IHL participants advised to also reverse the numerical order, so that the highest score would be underneath the strongest stream.

For the fifth question, 7 out of 16 AHL participants reported in the retrospective interviews that the face shown in front of the answer options does not contribute anything. Several times, it was even interpreted as one of the answer options.

During the interview, 3 out of 5 IHL participants and 1 out of 16 AHL participants recommended making the figures larger to reduce the interpretation problems. The only participant in the IHL group with a low command of Dutch remarked in the interview:

“Clear questionnaire, despite the fact that Dutch is not my mother tongue”.

### 3.2. Problems with Functional Tasks

#### 3.2.1. IPSS

*Generating* the required information did not appear to pose serious problems for participants, except for some difficulties mentioned by 4 out of 16 AHL participants and 1 out of 7 IHL participants in the retrospective interviews regarding the one-month time period that questions 1–7 refer to. One out of sixteen AHL participants noted while thinking aloud that the response options in the columns (from “never” to “almost always”) are not appropriate for question 7 (“How many times did you have to get up each night to pee during the past month?”). This problem led to a *verification* problem for this participant; he hesitated which answer to give. From what the participant said while thinking aloud, it appeared that this patient eventually did answer the question correctly. *Transforming* the requested information into a code shown in the table sometimes caused difficulties. Two out of sixteen AHL participants gave two answers to the utQOL question. In the interviews, they made it clear that they had not understood that this was not the intention.

#### 3.2.2. VPSS

After performing the interpretation tasks had led to correct interpretations of the questions*, generating* the requested information was not perceived as difficult. However, *verifying* in which category the information to be provided would fit presented some difficulties. In the third question, 2 out of 15 AHL participants reported during the interview that the urinary flow varies in strength over the day and that, therefore, one answer option was not sufficient for their complaints. One out of fifteen AHL participants noted while thinking aloud that there is no answer option for “no nycturia” in the second question. *Transforming* answers into codes went almost flawlessly, with one exception. One out of fifteen AHL participants said while thinking aloud that he believed the numbers mentioned in the third question actually belonged to the fourth question. Therefore, he did not answer the third question, the interview revealed.

#### 3.2.3. SVPI

*Generating* the required information did not pose any problems for the participants from both groups. Because the first option for least urgency in the third question was excluded by some participants as a possible answer (see Section 3.1.3), *verification* problems did arise. As a consequence, two out of sixteen AHL participants gave the wrong answer here, as the retrospective interviews showed. A transformation problem apparently arose with 1 out of 16 AHL participants who reported during the interview that he had actually wanted to tick the strongest urinary stream but in a rush had ticked the weakest stream.

### 3.3. Problems with Monitor Tasks

#### 3.3.1. IPSS

*Orientation problems* occurred with one out of seven IHL participants who overlooked the quality of life question. Five out of sixteen participants with AHL and two out of seven participants with IHL apparently encountered a *selectio**n* problem; the quality of life question was not answered due to the placement of the question in the questionnaire. For example, an AHL participant reported in the interview:

“Not so clear that this question belongs there too, because it is separate from the table with all other questions.”

Six out of sixteen AHL participants and one out of seven IHL participants recommended in the interview that this question be included in the table or that the text be framed, as it would then be clearer that the question belongs to the questionnaire.

The results of the form-filling activities were hardly ever *checked*. There was one out of seven IHL participants who read and answered question 3, then read the question again and changed the answer. None of the other participants checked their answers, as far as could be observed. *Switching* between interpretive and functional levels posed a problem for one out of seven IHL participants. This participant read question 5, crossed off the answer to question 6 and continued reading question 7. Because he was reading aloud, it was clear that he had not taken the time to read question 6 between questions 5 and 7.

#### 3.3.2. VPSS

The VPSS produced few problems in the monitor tasks. *Orientation problems* and *selection problems* did not occur. One out of fifteen AHL participants forgot to *switch* between the interpretation and the functional level when dealing with the third question. He read the question, worded his answer aloud but did not fill out this answer. When he *checked* the form after completing the last question, he noted the unanswered question. Only then did he fill out the correct answer.

#### 3.3.3. SVPI

No *orientation* or *selection* problems occurred while filling out the SVPI. None of the participants *checked* their answers. *Switching* problems were not observed.

### 3.4. Results from the Retrospective Interviews

After the participants had completed a form while thinking aloud, six standardized questions were asked during the retrospective interview (see Section 2.3). The first two questions asked the participants about their thoughts on the questionnaire and about their opinions on the layout. The answers helped in the interpretation of the participants’ utterances while thinking aloud, as presented in Section 3.1, Section 3.2 and Section 3.3. The results of the remaining four questions are presented below.

The third question was: “Is the questionnaire more difficult or easier compared to other forms you are familiar with?”. Here, the participants could choose from four answer options: “easier”, “equal”, “more difficult” and “don’t know”. It appeared that most AHL participants (9 out of 16) found the IPSS easier than other forms with which they were familiar. For the IHL participants who had completed this form, it was different. Most of them (four out of seven) found the IPSS more difficult than other forms. For the VPSS, there were as many AHL participants (6 out of 15) who found it easier than other forms as AHL participants who found it more difficult; this was also true for the IHL participants who had completed the VPSS (both times 3 out of 6). The SVPI was found easier than other forms known to them by almost all participants in the AHL group (13 out of 16) and by all (5) participants in the IHL group.

The fourth question was: “Did you miss any general information, such as an introduction at the top of the questionnaire?”. In both the IPSS and the VPSS, a minority of AHL participants indicated that they had missed general information (3 out of 16 and 4 out of 15, respectively). In both these forms, this was also true for a majority of IHL participants (five out of seven and five out of six, respectively). Participants who had filled out the SVPI only seldomly reported that they had missed general information (1 out of 16 in the AHL group and none out of 5 in the IHL group).

The fifth question was an open-ended question: “How did you experience this test?”. Here, 17 out of 18 participants in the IHL group responded positively (“useful”, “easy” or “good”, for instance). The IHL participant who responded negatively had completed the IPSS and said: “The recording is a little disturbing; I already have difficulty reading and a recording is unpleasant”. All 47 AHL participants responded positively to this question, with answers such as “nice”, “fine”, “pleasant” and “useful”.

The sixth and final question was: “Should you have completed this questionnaire for yourself at home, would you have proceeded in the same way as you have done now?”. Again, 17 out of 18 IHL participants agreed. The IHL participant who responded negatively was the same participant quoted in the previous question. On the sixth question, he said: “I would have answered the same, only I would have filled out questions that I have now left open, not knowing if my answers would have been correct”. All 47 AHL participants answered “yes” to this question.

## 4. Discussion

With this qualitative study, we attempted to gain more insight into the nature of problems that men face in completing an existing LUTS questionnaire. Furthermore, we wanted to identify the underlying causes for these problems, and we intended to determine the association with the level of health literacy. Below, we discuss our main findings and the relationship between our results and the results of previous studies.

### 4.1. Problems of Various Nature

Analysis of the LUTS questionnaires completed by the participants (a total of 65 men from the target group), of the interviews we conducted with the participants, of the behavioral observations during the completion and, in particular, of the participants’ thinking aloud utterances, revealed serious problems of various nature, corresponding with different subtasks that have to be performed during the completion of the questionnaires. Semantic and pragmatic interpretation tasks proved especially difficult. Most noticeable were the interpretation problems with the IPSS, the so-called gold standard among the LUTS questionnaires.

Participants did not always grasp the meaning of the terms and pictograms used in the questionnaires (*semantic interpretation*), and sometimes, they did not understand what actions were expected of them (*pragmatic interpretation*). In the IPSS, the terminology led to many interpretation problems. The table in this questionnaire also caused difficulties for some of the participants because they did not make the connection between the numbers in the cells and the answer options in the top row. In both the VPSS and SVPI, there were pictograms that caused interpretation problems.

*Functional tasks*, such as categorizing one’s own situation (*verifying*) and converting information into a code used in the questionnaire (*transforming*), also did not always prove easy to perform. As with the interpretation tasks, most of the problems here occurred in the IPSS, and particularly in filling out the table and in answering the question below it. In the other two questionnaires, the functional tasks presented fewer difficulties.

Finally, there were *monitoring tasks*, in particular *orientation* in the questionnaire and *selection* of the relevant questions, which posed problems. Again, the IPSS proved difficult for a number of participants. With the VPSS and the SVPI, few, if any, orientation problems were noted.

### 4.2. Underlying Causes

Several probable causes were identified for the completion problems observed. In the IPSS, various terms are used, which deviate from the everyday language of the form fillers. This questionnaire also showed that some of the participants were apparently not sufficiently accustomed to working with tables. Finally, the layout of the IPSS proved to be problematic. In the VPSS, some of the pictograms, partly due to the lack of verbal support, did not match the prior knowledge of a number of participants. Additionally, in the SVPI, there was sometimes a discrepancy between the pictograms used and the prior knowledge of the form fillers. Additionally, the use of a numerical scale in one of the questions deviated from what would have been logical in the eyes of the participants. The retrospective interviews revealed that compared to other forms that participants were familiar with, the ease of completion of the SVPI in particular was rated higher. However, it was clear from the thinking-aloud utterances that despite the verbal support in this questionnaire, the pictograms in one of the questions led to serious interpretation problems.

### 4.3. Differences between Groups with Various Levels of Health Literacy

When completing the IPSS, participants with an inadequate level of health literacy (IHL) had more problems with the terminology used than those with an adequate level of health literacy (AHL). However, an adequate level of health literacy certainly did not guarantee error-free and effortless completion of the table that is central in this questionnaire. It also appeared that selection problems were not limited to participants in the IHL group. The last question in this questionnaire was also overlooked by a number of AHL participants. In the VPSS, there were no clear health-literacy-related differences in the frequencies of the problems. What was striking here was that AHL participants often understood what was meant with the first question after seeing the second question. Such subsequent understanding did not occur among IHL participants. In the SVPI, both AHL and IHL participants experienced problems understanding the pictograms in one of the questions. Confusion due to the number scale used was only observed with participants in the AHL group. All in all, the results of the three questionnaires make it clear that they may lead to problems for form fillers in both the IHL and the AHL group.

### 4.4. Relationship with the Results of Previous Studies

Previous quantitative studies into the American IPSS, the most frequently used LUTS questionnaire, showed that participants frequently provided incorrect information.

Earlier quantitative studies also revealed that the VPSS, the visual alternative to the IPSS developed in South Africa, is difficult to complete for part of the patients. Previous quantitative research has not yet provided a clear picture of the difficulty of the SVPI, the French alternative to the IPSS.

Until now, it was unknown to which causes the observed completion problems with LUTS questionnaires should be attributed. The present qualitative study shows where the main cause of the difficulties lies: all three questionnaires contain elements that presuppose knowledge or skills that part of the participants do not possess.

Previous qualitative research on form-filling behavior that inspired the method used in this study dealt with problems that Dutch participants experienced when they filled out government forms. Serious problems were found in almost every task and subtask the fillers had to perform [24,27] (see Figure 4). In our study, this was also the case with the IPSS. With the visual alternatives, VPSS and SVPI, the problems occurred mainly in the interpretation tasks.

The causes mentioned by Jansen and Steehouder [24,27] for the problems they identified are also recognizable in our study. Often, the form designers had apparently overestimated the *prior knowledge* of the filler. In the case of the IPSS, this concerned the terminology used; in the case of the VPSS and IPSS, it concerned some of the pictograms used. In a number of cases, the *skills* of the fillers had also been apparently overestimated. In the case of the IPSS, this concerns the use of a table; in the case of the SVPI, it concerns interpreting a scale. Sometimes, but less often than with the longer and more complicated forms used in the Jansen and Steehouder study, the *willingness* to work through the entire form was overestimated. However, in the IPSS, the last question was often overlooked. Here, the poor layout of this form did not seem to fit with the perhaps too hasty attitude of some participants.

### 4.5. Strengths and Limitations

To our knowledge, this is the first qualitative study into problems encountered by men when completing a LUTS questionnaire. It provides new insights into how men go about filling in the IPSS and its visual alternatives, the VPSS and the SVPI, and it suggests opportunities for developing a new and improved LUTS questionnaire.

The first and most important strength of this study is that by applying the working-aloud method, we were able to gain a level of understanding of the barriers to correctly completing these questionnaires that quantitative research cannot provide. Second, the distinction we were able to make between groups of participants with different levels of health literacy allowed us to discover that the completion problems with LUTS forms are not limited to men with low levels of health literacy. We were able to conclude that the terminology in the IPSS was especially difficult for men with an inadequate level of health literacy. However, other types of problems with the IPSS, as well as interpretation problems with the other forms, also proved to occur among men with an adequate level of health literacy.

This study also has some limitations. The qualitative nature of the study and the limitation of the participants to a sample of no more than 65 men in the Netherlands imply that the findings should not be generalized but rather considered as a valuable source of ideas for possible improvement of LUTS questionnaires. It should be noted, however, that also in other studies on symptom assessment instruments using the thinking aloud method, only limited numbers of participants are reported. For instance, a Canadian and a Norwegian study into the *Edmonton Symptom Assessment System* (ESAS), a tool asking advanced cancer patients for self-reports of symptom intensity, included only 20 and 11 participants, respectively. In these studies, it was found that despite the fact that a majority of patients stated that the ESAS is easy to complete, they struggled with interpreting some of the symptoms and numerical scales, and that errors occurred regarding the interpretation of the symptoms [36,37].

It would have been better if we could have recruited all participants in one round in an outpatient clinic. Because we had to hold a second round outside the outpatient clinic to recruit more IHL participants, it cannot be excluded that the AHL participants in this study were, on average, more familiar with LUTS complaints than the IHL participants, with the possible consequence that the AHL participants recognized more elements from questionnaires than the IHL participants. Furthermore, only the IHL group comprised non-native speakers of Dutch: 4 out of 18. Moreover, the AHL participants were, on average, older than the IHL participants (65 and 61 years, respectively). As LUTS tend to increase with age, this may have also led to some bias in the outcomes. However, the results we found do not indicate a great effect of these circumstances. AHL participants also sometimes encountered serious obstacles in completing the questionnaire, and all three questionnaires appeared to lead to problems for form fillers in both the IHL and the AHL group.

In addition, although care was taken to ensure that the information that the test leader could deduce about the health literacy status of the participant would not affect the continuation of the session, such influence cannot be completely excluded.

As in other research where the thinking aloud technique is used, participants in this study also evidently did not verbalize all their thoughts. As always, there were silences and verbal fillers, such as “um” and “ah”, which are difficult to interpret. Therefore, it is often recommended that the use of multiple methods be used to increase the validity of the study and to gain more insight into elements that contribute to the cognitive-processing difficulty [32,38]. This is what we implemented in this study by applying the working-aloud method, in which we combined the analysis of thinking aloud utterances with behavioral observation, analysis of completed forms and information provided by participants in retrospective interviews.

### 4.6. Implications

In view of the shortcomings of the IPSS found in previous quantitative research, the status of this questionnaire as the gold standard has already been questioned (in Refs. [11,14,16], for instance). Our qualitative research added such obvious shortcomings of the IPSS that there is a clear need for better alternatives. Such alternatives could perhaps be the VPSS and SVPI. However, we found that even these questionnaires may present serious completion problems for part of the patients. It is therefore advisable that a new LUTS questionnaire be developed that reduces the completion problems for both men with an inadequate and an adequate level of health literacy, and whose results show strong positive correlations with objective flowmetry measurements.

In the development of such a new LUTS questionnaire, the findings from this qualitative study could play an important role. Given the possible interpretation problems of terminology that may not be common to patients with inadequate literacy, the use of pictograms in a new LUTS questionnaire is an obvious choice. However, publications on the effects of pictograms in a medical context, in particular, suggest that it would be wise to combine the pictograms with short simple texts [39,40,41,42]. The comprehensibility and ease of completion of such a new LUTS questionnaire should also be thoroughly tested in a qualitative study, preferably using the working-aloud method [27,28,29,32]. Afterward, the questionnaire, possibly further adapted, can be validated in a quantitative study and, if the results of such a study are positive, put into use.

## 5. Conclusions

Each of the LUTS questionnaires examined in this study has been found to cause problems for both men with an inadequate and an adequate level of health literacy. Overestimation of patients’ knowledge and skills by forms designers can be identified as a major explanation of the problems experienced by form fillers. As this study shows, the working-aloud method with different groups of participants can provide insights into form-filling behavior that quantitative research cannot. The insights gained in this study can be used as a basis for a new LUTS questionnaire, which may lead to fewer completion problems for patients and provide urologists with more reliable information.

## Figures and Tables

**Figure 1 ijerph-19-09650-f001:**
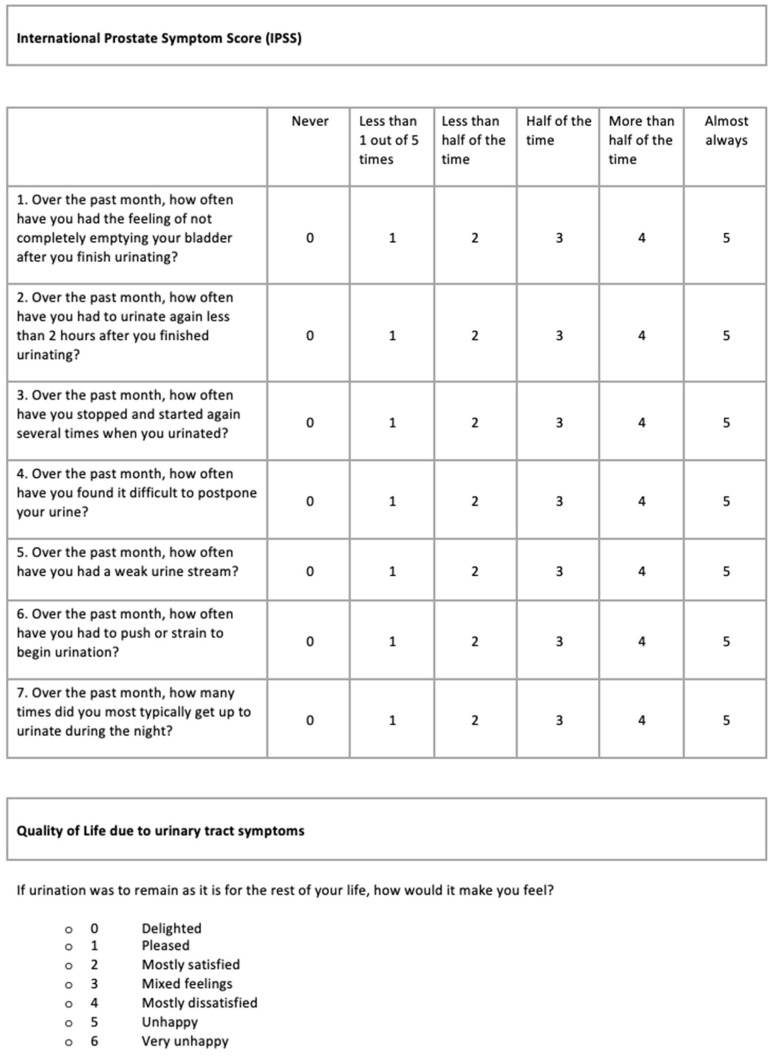
International Prostate Symptom Score (IPSS). Dutch translations: “Internationale Prostaat Symptonen Score (IPSS)”. “Hoe vaak had u de afgelopen maand het gevoel dat de blaas nog niet leeg was nadat u had geplast?”. “Nooit”; “Minder dan 1 van de 5 keer”; “Minder dan de helft van de keren”; “De helft van de keren”; “Meer dan de helft van de keren”; “Bijna altijd”. “Hoe vaak moest u de afgelopen maand binnen 2 uur nadat u geplast had weer plassen?”. “Hoe vaak merkte u de afgelopen maand dat tijdens het plassen de straal enkele keren stopte en weer begon?”. “Hoe vaak had u de afgelopen maand moeite om het plassen uit te stellen?”. “Hoe vaak had u de afgelopen maand een zwakke urinestraal?”. “Hoe vaak moest u de afgelopen maand persen om de urinestraal op gang te brengen?”. “Hoe vaak moest u de afgelopen maand per nacht opstaan om te plassen?”. “Quality of Life ten gevolge van symptomen van uriwegen”. “Als het plassen uw hele leven zou blijven zoals het nu is, hoe zou u zich daar bij voelen?”. “Gelukkig”; “Plezierig”; “Over het algemeen tevreden”; “Gemengde gevoelens”; “Over het algemeen ontevreden”; “Ongelukkig”; “Zeer ongelukkig”.

**Figure 2 ijerph-19-09650-f002:**
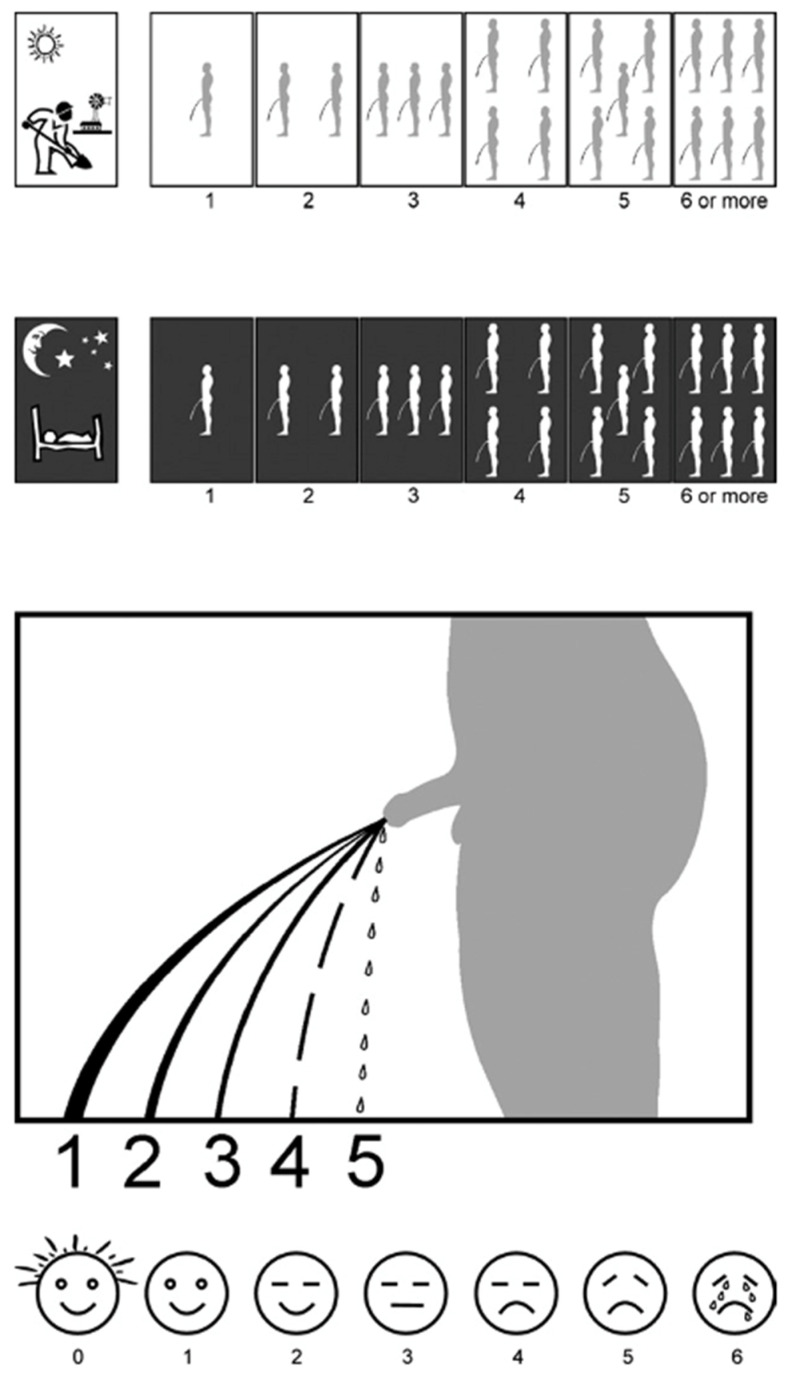
Visual Prostate Symptom Score (VPSS) [9]. Dutch translation of “6 or more”: “6 of meer”.

**Figure 3 ijerph-19-09650-f003:**
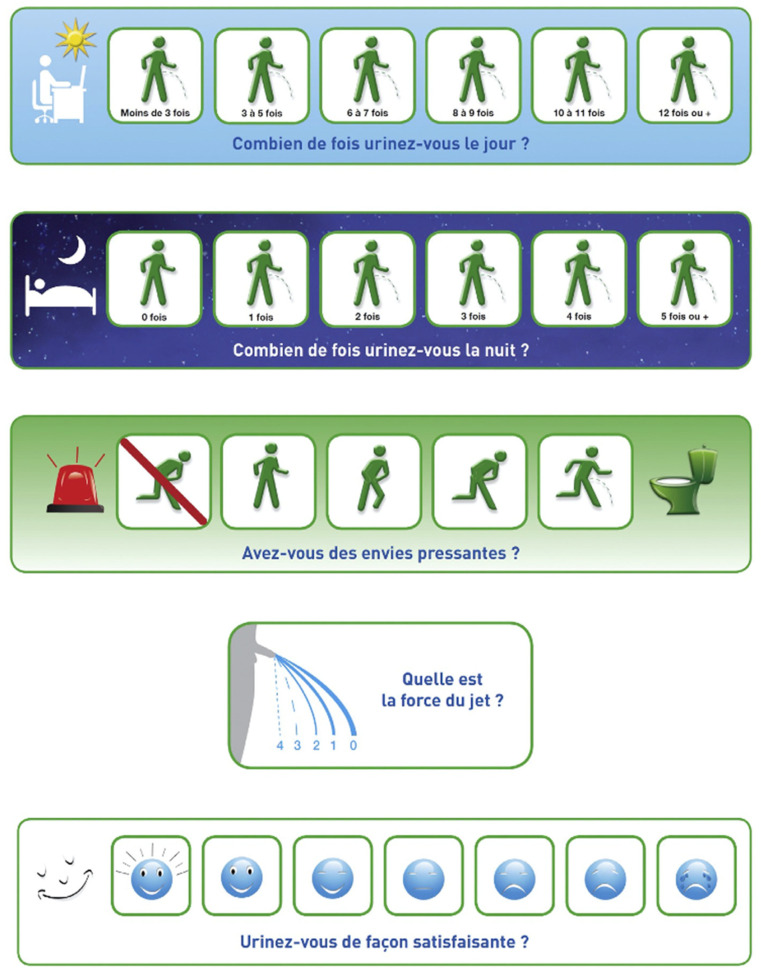
Score Visuel Prostatique en Image (SVPI) [21]. English translations: “Less than 3 times”; “3 to 5 times”; “6 to 7 times”; “8 to 9 times”; “10 to 11 times”; “12 times or +”. “How often do you urinate during the day?”. “0 times”; “1 time”; “2 times”, “3 times”; “4 times”; “5 times or +”. “How often do you urinate at night?”. “Do you have pressing urges?”. “How strong is the urine stream?”. “Do you urinate satisfactorily?”. Dutch translations: “Minder dan 3 keer”; ”3 tot 5 keer”; “6 of 7 keer”; “8 of 9 keer”; “10 of 11 keer”; “12 keer of meer”. “Hoe vaak moet u overdag plassen?”. “0 keer”; “1 keer”; “2 keer”, “3 keer”; “4 keer”; “5 keer of meer”. “Hoe vaak moet u ‘s nachts plassen?”. “Hebt u wel eens hoge nood?”. “Wat is de kracht van de straal?”. “Hoe tevreden bent u over het plassen?”.

**Figure 4 ijerph-19-09650-f004:**
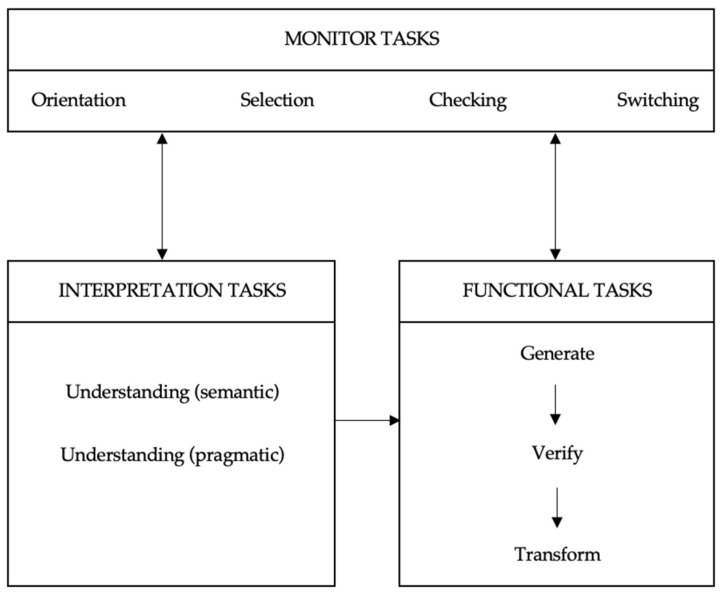
Task analysis.

**Table 1 ijerph-19-09650-t001:** Participant characteristics.

	IPSS	VPSS	SVPI	Total
Participants (n)				
AHL	16	15	16	47
IHL	7	6	5	18
Total	23	21	21	65
Med. age (years)				
AHL	75.5 (48–82)	75 (43–84)	65 (40–78)	65 (40–84)
IHL	52 (41–64)	61.5 (57–65)	64 (61–68)	61 (41–68)
Total	72 (41–82)	69 (43–85)	65 (40–78)	65 (40–85)
Language				
AHL				
(near) native	16	15	16	47
non-native	0	0	0	0
IHL				
(near) native	5	5	4	14
non-native	2	1	1	4
Total				
(near) native	21	20	20	61
non-native	2	1	1	4

## Data Availability

The data presented in this study are available on request from the corresponding author.

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
