# Peer review of "How Do Patients Understand Questions about Lower Urinary Tract Symptoms? A Qualitative Study of Problems in Completing Urological Questionnaires"

_ijerph, 2022, doi:10.3390/ijerph19159650_

Round 1
Reviewer 1 Report
The article is devoted to the actual theme - the information search for an effective and reliable tool to assess the lower urinary tract symptoms (LUTS) in men. The scientific work deals with important issues of qualitative filling of the existing generally accepted the International Prostate Symptom Score (IPSS) questionnaire and its existing alternatives for people with different levels of general education, including health literacy.
This article is also of great practical importance. In real medical practice, it is often necessary to face a situation when patients do not correctly understand IPSS questions and cannot give adequate answers to them. This is also important because most algorithms for differential diagnosis of LUTS are based precisely on the assessment of the nature and severity of symptoms.
The object of the study were three questionnaires designed to assess the symptoms of impaired urination: the common IPSS, Visual Prostate Symptom Score (VPSS) and the combined questionnaire with visualization and explanations in a summary - Score Visuel Prostatique en Image (SVPI).
The research included 65 men who had not previously filled out any of the listed questionnaires. To specifying the level of health literacy, all participants completed the Measuring level of health literacy (SBSQ). After that all the participants were divided into groups with adequate health literacy (AHL) and inadequate health literacy (IHL). Difficulties encountered in completing the questionnaires can be divided into three main groups: semantic interpretation, pragmatic interpretation, functional tasks.
As a result of the study, the authors received in-depth understanding of the reasons for incomplete or unreliable filling of this questionnaires. Most noticeable were problems with the IPSS. In analyzing the results, significant difficulties were identified in the interpretation of the terminology used in the IPSS. Difficult for understanding were also the ratio of questions and answers in the IPSS table. The alternative questionnaires (VPSS and SVPI) also showed significant difficulties for patients in visualizing their own complaints. The study demonstrated a significant predominance of IPSS filling quality in patients with AHL. However, AHL did not guarantee a reliable 100% completion of this questionnaire. Significant difference between the quality of filling out VPSS and SVPI in patients with different levels of the healthy literacy was not found.
The F. Schlatmann’s et al. study revealed a number of difficulties in filling out the IPSS, which raise doubts about the effectiveness of this questionnaire, which was known from previous studies. However, the alternative questionnaires (VPSS and SVPI) are also not perfect. Thus a new questionnaire needs to be developed, which would meet all the necessary requirements, and would not cause any difficulties in filling it out and significantly increase the effectiveness of assessing the severity LUTS, regardless of the level of health literacy.
Summing up, we can conclude that the article has a certain scientific and practical value.

Reviewer 2 Report
Given that symptom scores are used so frequently, I feel that the present work questioning their validity is relevant. My main concerns relate to issues with reporting and an insufficient discussion of potential biases in this study. While the number of comments is large, I consider all of them fixable by better explanations and discussion.
Main comments:
1. I did not fully understand the basic study design based on the Abstract. Apparently, three questionnaires were studied and men “filled out one of these questionnaires”. Does that mean that each participant filled only one of the three questionnaires, implying the samples size per questionnaire was not 18/47 but more like 6/16?
2. Section 1.2.1: I admit that I was not aware of the VPSS, which possibly is true for many readers. Therefore, it may be helpful if the 1st paragraph in this section explains more how this score works. Figure 2 helps with that, but some of that information should already be in the Introduction.
3. While I understand the self-imposed study questions as relevant, the preceding description of existing data on the three scores does not make it obvious why the authors’ questions address a knowledge gap. I would find it helpful if the preceding text not only states problems with the three scores but already identifies knowledge gaps that the present study attempts to address.
4. It did not become clear to me from section 2.2 whether presence of any LUTS was an inclusion criterion of the study. However, this seems important to me as certain questions may be less obvious to men without LUTS. Thus, please add an explicit statement whether presence of any LUTS was an inclusion criterion or not. That participants attended an outpatient clinic does not substitute because that technically could be men with fertility issues, cancers or other conditions not necessarily implying LUTS. While I understand the need for a 2nd round recruitment for the IHL group, this causes biases that need to be discussed. If the IHL group includes more men without LUTS, it could be that poorer understanding of the scores is not only a question of health literacy but also of men without LUTS having more problems filling the scores than those with LUTS. For instance, explaining urgency to a man who never had it is not so easy.
5. The overall Methods section is insufficiently clear on sample size and allocation. A) Was the study based on a pre-specified number of men in each group? If not, were sample sizes adapted after results had been seen (something known to cause major bias)? B) How were participants allocated to the three questionnaires? Some kind of randomization or a different method that may have introduced biases?
6. The Methods section should explicitly state whether the interviewer was aware of the IHL/AHL status of the participant because such knowledge may bias the qualitative assessment of their statements. The implications of such bias should be discussed.
7. Were the utterings of the participants recorded for further analysis beyond the personal recollection of the interviewer?
8. Table 1: I notice that the median age in the IHL groups were markedly lower than in the AHL group (>20 years in the IPSS cohort). As LUTS tend to increase with age, possibly personal LUTS experience was less in IHL groups simply because they had less experience with it. This must be discussed as a specific limitation of the study.
9. Another issue apparent from Table 1 is that the fraction of native speakers in the AHL groups was much higher than in the IHL groups. Thus, I wonder how much of the perceived poor health literacy is really that and how much is simply not being good at speaking/reading Dutch. This should be discussed.
10. Section 3.1: As the total group sizes for AHL were >2-fold as for IHL, the absolute numbers in this section are not easily interpreted. For instance, “4 AHL … and 4 IHL” in l. 471 creates an impression that the problem was equally present in both groups. However, if it would read “4/16 AHL … and 4/7 IHL” an impression would be created that may be closer to the true observations. I agree that reporting absolute numbers is appropriate in qualitative studies with small sample sizes. However, if sample sizes substantially differ between groups to be compared, reporting them relative to group size may be helpful to avoid misleading impressions.
11. I personally am surprised that none of the participant had an issue with the reversal of axes for question 8 vs. 1-7. If my impression is true, that may be worth noting explicitly.
12. Section 4 is largely free of references. Should I interpret this as saying that this research was so innovative that we could not compare our findings to anything else? If true, it may be helpful to compare present findings to those from qualitative studies on symptom scores outside of urology. Since some of the authors are not urologists, they can perhaps think of fields to which the present data can be compared.
13. I found section 4.5 (strengths and limitations) to be too short and missing important aspects. I personally see two key limitations of the current study that are insufficiently presented here. Firstly, the used sample sizes (I do not even know from Methods whether these had been planned to be that low) create a major risk of sampling errors, particularly in the IHL group. Second, I have noted several sources of possible bias in my preceding comments. I strongly recommend that their presence and potential implications are discussed here.
14. Appendix A: I found this table difficult to interpret since a key piece of information is missing. Did the authors of the underlying studies use parametric or non-parametric tests for r? While many studies have used parametric tests for r, a lot of research has shown that this is inherently flawed because the underlying parameters exhibit a distribution considerably deviating from a normal distribution. As a bare minimum, such information should be added to the legend. Moreover, the authors should add sample sizes, perhaps as part of the left column.
Other comments:
15. L. 13: do you mean “ageing” (everybody ages starting immediately after birth) or “aged”?
16. Abstract and main Introduction state that LUTS are common in men. While this is true, it is equally true for women – but that is not the impression that is created by the present wording. Of note, nothing wrong in doing this study in men only.
17. L. 55: I assume that the “p. 626” refers to a specific place in reference #10. When looking at the Johnson paper, the overtreatment is a speculation coming from these authors, not a finding they report. I feel that it should become clearer that reference #10 did not show this but only speculated in this regard.
18. L. 59: “decided to design” is unnecessarily complicated. They not only decided but actually did it. Thus, just saying “designed” may be sufficient; this also applies to l. 88.
19. It is odd that the abbreviation VPSS is introduced in l. 61, but first used in the header in l. 56.
20. L. 76: When the authors say “variance”, I assume they think of the plain English meaning of the word, which is more or less synonymous with “variability”. However, they also talk about r, implying some type of statistical analysis. Within statistics, variance is typically defined as SD squared – a meaning apparently not intended here. Therefore, I suggest rewording this sentence. Scientifically more relevant is the “explained” in l. 76. Correlations show just that, but never imply a cause-effect relationship. Thus, they intrinsically cannot explain something; a more appropriate wording may be “mathematically attributed”.
21. L. 91: I feel that the “significant correlations” is unnecessarily ambiguous. “significant” means important, relevant or similar in plain English, but an observed p-values smaller than the pre-specified statistical alpha in a statistical context. An important effect and one with a low p-value are not necessarily associated unless sample size is large. Therefore, I suggest replacing the “significant” here with what you really wish to convey.
22. The text from l. 97 onward technically is part of section 1.2.2, but intellectually does not belong to that section because it is not specifically related to SVPI. Should there be a new subheading indicating that the subsequent text will describe the study question?
23. L. 127-129 represent a duplicated statement.
24. The information on the Brazilian score (l. 179-182) is unrelated to Methods and accordingly does not belong here. It could become part of section 1 or 4 but could also be removed.
25. L. 183-185: A major reason for the global popularity of the IPSS is that it has been translated into many languages with formal validation studies of each of those translations. Thus, please clarify whether the Dutch version of the IPSS has undergone formal language validation or is a homemade product of the mentioned hospital in Arnhem.
26. L. 184-196 extensively describe how the Dutch version VPSS and SVPI were created (good!). However, it would be helpful to at least the readers understanding Dutch if those final versions were presented in Dutch, perhaps as part of the supplement.
27. L. 198-200: I consider this statement wrong. While often misinterpreted that way, the IPSS was neither designed nor validated to aid in establishing a diagnosis; it primarily is a tool to quantify symptoms (despite the “prostate” in its name, IPSS scores in women can be as high as in men). Thus, the IPSS “can also be used for monitoring” is not an “also” thing but the primary purpose of this score (and the VPSS and SVPI).
28. L. 674: “not nearly always” sounds clumsy to me.
29. L. 834: This is an incomplete sentence that does not make sense in its present form.
Reviewer 3 Report
Dear Authors,
The manuscript is interesting and compares the understanding in the general population of three questionnaires used in LUTS. The impact of understanding the questions and providing the answers plays a vital role in the therapy decision. This manuscript could serve to improve the urological questionnaires.
The article is well written and organized.
Round 2
Reviewer 2 Report
All of my previous comments have been addressed adequately.